# Seroepidemiology of *Toxoplasma gondii* infection in people with alcohol consumption in Durango, Mexico

**Sergio Estrada-Martinez**[1], **Alma Rosa Pérez-Álamos**[1], **Melina Ibarra-Segovia**[2],
**Isabel Beristaín-Garcia**[2], **Agar Ramos-Nevárez**[3], **Leandro Saenz-Soto**[3], **Elizabeth Rábago-Sánchez**[4], **Carlos Alberto Guido-Arreola**[3], **Cosme Alvarado-Esquivel**[4]*

**1** Institute for Scientific Research "Dr. Roberto Rivera-Damm", Juárez University of Durango State, Durango, Mexico, **2** Facultad de Enfermería y Obstetricia, Juárez University of Durango State, Durango, Mexico, **3** Clinical Laboratory, Clínica de Medicina Familiar, Instituto de Seguridad y Servicios Sociales de los Trabajadores del Estado, Durango, Mexico, **4** Biomedical Research Laboratory, Faculty of Medicine and Nutrition, Juárez University of Durango State, Durango, Mexico

* alvaradocosme@yahoo.com

**Data Availability Statement:** All relevant data are within the paper and its Supporting Information files.

## Abstract

The seroepidemiology of infection with *Toxoplasma gondii* (*T. gondii*) in alcohol consumers is largely undeveloped. In light of this, we sought to determine the seroprevalence of *T. gondii* infection in alcohol consumers in Durango, Mexico, and the association of *T. gondii* seroprevalence with characteristics of the population studied. Anti-*T. gondii* IgG and IgM antibodies were searched in sera of participants using commercially available enzyme immunoassays. Bivariate and logistic regression analyses were then used to determine the association between *T. gondii* infection and the characteristics of the population studied. Of the 1544 people studied (mean age: 39.4±14.0 years), 173 (11.2%) tested positive for anti-*T. gondii* IgG antibodies. We were able to test 167 of the 173 anti-*T. gondii* IgG positive sera for anti-*T. gondii* IgM antibodies. Fifty-five (32.9%) of these 167 serum samples were positive for anti-*T. gondii* IgM antibodies. Bivariate analysis showed that visual impairment, history of surgery, and hepatitis were negatively associated with *T. gondii* IgG seropositivity (*P*<0.05). In women, seropositivity to *T. gondii* was positively associated with a history of pregnancy (*P*<0.05). Logistic regression analysis showed that *T. gondii* seropositivity was associated with the variables consumption of armadillo meat (OR = 2.33; 95% CI: 1.04–5.22; *P* = 0.03), and the use of latrines for elimination of excretes (OR = 2.27; 95% CI: 1.07–4.80; *P* = 0.03); and high (>150 IU/ml) anti-*T. gondii* IgG antibodies were associated with consumption of both armadillo meat (OR = 2.25; 95% CI: 1.01–5.02; *P* = 0.04) and crowding at home (OR = 1.63; 95% CI: 1.02–2.61; *P* = 0.03). We found a distinct *T. gondii* seroprevalence in people with alcohol consumption from those previously found in population groups in the region. This is the first study that illustrates the association between high anti-*T. gondii* antibodies and crowding in Mexico, and the second study on the association between *T. gondii* infection and consumption of armadillo meat and the use of latrines in this country. We conclude that epidemiology of *T. gondii* infection in people with alcohol consumption deserves further investigation.

**Funding:** This study was financially supported by Universidad Juárez del Estado de Durango, Mexico. The funders had no role in study design, data collection and analysis, decision to publish, or preparation of the manuscript.

**Competing interests:** The authors have declared that no competing interests exist.

## Introduction

The parasite *Toxoplasma gondii* (*T. gondii*) is a zoonotic pathogen belonging to apicomplexan parasites [1]. Infections of this parasite occur widely around the world [2]. It is estimated that about 30% of people are seropositive to *T. gondii* worldwide [3]. Cats and other Felidae are the definitive hosts of *T. gondii*, whereas a wide variety of warm-blooded animals, including humans, are intermediate hosts [4]. Humans can become infected with *T. gondii* through ingestion of oocyst-contaminated soil and water, tissue cysts in undercooked meat, or congenitally [5]. Infections with *T. gondii* are usually asymptomatic, but the parasite may induce severe disease in fetuses and immunocompromised patients [6]. Furthermore, *T. gondii* can cause posterior uveitis with vision loss [7]. Infection with *T. gondii* has also been linked to an increased incidence of several psychiatric diseases [8]. Chronic infections with *T. gondii* have been more recently linked to behavioral changes [9]. Suicide attempters have shown a higher seroprevalence of *T. gondii* infection than healthy controls [10]. In addition, infections with *T. gondii* have been linked to traffic [11, 12] and work accidents [13].

To the best of our knowledge, there have not been any studies done on the seroepidemiology of *T. gondii* infection in people with alcohol consumption. We believe it is important to study the seroepidemiology of *T. gondii* infection in this population group because alcohol consumption has been linked to infection with *T. gondii* in several studies. Seropositivity to *T. gondii* was associated with alcohol consumption in patients with heart disease [14]. In two studies of decedents in Poland, researchers found a positive association between *T. gondii* seropositivity and the presence of alcohol in the blood [15, 16]. In addition, alcohol consumption leads to an impaired immune response and dysregulated inflammatory state that contributes to an increased risk for infection [17]. On the other hand, infection with *T. gondii* has been associated with depression [18, 19], an illness that may lead to alcohol consumption. Therefore, we see a need to study the magnitude of infection with *T. gondii* and the factors associated with this infection in people with alcohol consumption. In this study, we sought to determine the seroprevalence of *T. gondii* infection in alcohol consumers in the northern Mexican city of Durango, and the *T. gondii* seroprevalence association with sociodemographic, clinical, and behavioral characteristics of the population studied.

## Materials and methods

### Study design and study population

We performed a cross sectional study from June 2014 to May 2018 in which we surveyed 1544 people with alcohol consumption in Durango State, Mexico. The inclusion criteria for enrollment of participants in the survey were: alcohol consumers, aged 15 years and older, and willing to participate in the study. "Alcohol consumption" is defined as the act of ingesting -typically orally- a beverage containing ethanol [20]. We used this criterium for enrollment of participants; however, we further considered quantity and frequency of alcohol drinking. Thereafter, alcohol consumption was defined as: consumption of any alcoholic drink (beer, wine, tequila, brandy, vodka, whisky, etc.) at least once a month during the previous six months. Occupation, gender, residence, education, socioeconomic status, or presence of any disease were not restrictive criteria for enrollment. Exclusion criteria were insufficient blood sample or unwillingness to provide information. Participants were enrolled in 9 public hospitals and health care centers in urban and rural areas of Durango when attending for health check-ups or medical consultations or when health care providers visit them at their workplace or home. Patients in these settings were informed about the study and invited to participate. Only patients that fulfilled the inclusion criteria were enrolled in the study.

## Data collection and socio-demographic, clinical, behavioral, and housing variables of participants

A standardized questionnaire was administered to participants concerning their socio-demographic, clinical, behavioral, and housing characteristics. We saw no need to perform a validation of the questionnaire because we did not attempt to measure a variable or make a diagnosis with it. The questionnaire was used as a method for recording simple variables as age, gender, etc. The questionnaire has been used in a number of epidemiological studies in the same language in the same country [13, 14, 18, 21–26]. Items of housing variables were previously validated [27]. We obtained information about the age, gender, birthplace, residence, education, occupation, and socioeconomic status of participants. Clinical data included presence of any disease, dizziness, frequent abdominal pain or headache, impairments in memory, reflexes, hearing and vision, and history of blood transfusion, organ transplantation, lymphadenopathy, hepatitis, or surgery. Obstetric history in women was also obtained. With respect to behavioral data, we recorded information about contact with soil, cleaning cat feces, contact with cats and other animals, travel (in Mexico or abroad), frequency of eating away from home (in restaurants or fast-food outlets), washing hands before eating, consumption of unpasteurized milk, untreated water, unwashed raw vegetables or fruits, type of meat consumed, frequency of meat consumption, eating raw or undercooked meat, animal brains, dried or cured meat, or beef liver. We also recorded history of drug use, tobacco consumption, sexual promiscuity, and the following housing conditions: crowding, availability of potable water, type of flooring, form of elimination of excretes, and education of the head of the family.

## Detection of anti-*T. gondii* IgG and IgM antibodies

A serum sample from each participant was obtained and stored at –20˚C until analyzed. Detection and quantification of anti-*T. gondii* IgG antibodies were performed using a commercially available enzyme immunoassay "*Toxoplasma gondii* IgG" kit (Diagnostic Automation/Cortez Diagnostics, Inc., Woodland Hills, California. USA). Detection of anti-*T. gondii* IgM antibodies was performed using a commercially available enzyme immunoassay "*Toxoplasma gondii* IgM" kit (Diagnostic Automation/Cortez Diagnostics, Inc.). Only sera with seroreactivity to *T. gondii* IgG were further tested for IgM antibodies. Positive and negative controls were included in each run. Both IgG and IgM tests were performed following the instructions of the manufacturer. The immune status ratio (ISR or index) for anti-*T. gondii* IgG and IgM antibodies were calculated by dividing the sample optical density value by the cut-off calibrator value. We considered the sample positive for IgG or IgM when an IgG index or an IgM index was $\geq$1.1. Seropositivity was a qualitative measurement and considered the presence of anti-*T. gondii* IgG antibodies (regardless the levels), whereas serointensity was a quantitative measurement and considered the serum levels of anti-*T. gondii* IgG antibodies expressed as international units per milliliter (IU/ml). High levels ($>$150 IU/ml) of anti-*T. gondii* IgG antibodies were considered as a strong antibody response probably due to continuous exposure to the parasite.

## Statistical analysis

Data analysis was performed with the aid of the software EPIDAT 3.1 and SPSS version 15.0 (SPSS Inc. Chicago, IL. USA). We calculated the sample size (n = 1543) based on the following parameters: a 6.1% estimated seroprevalence of *T. gondii* infection [21], 336,606 as the size of population from which the sample was selected, a precision of 1%, and a confidence level of 90%. We estimate there to be 336,606 alcohol consumers in the state. We arrived at this

number by extrapolating the 51.4% prevalence of alcohol consumption reported in Mexico (https://encuestas.insp.mx/ena/ena2011/factsheet_alcohol25oct.pdf) from the number (654,876) of inhabitants in the municipality of Durango (http://cuentame.inegi.org.mx/monografias/informacion/dur/poblacion/default.aspx). To compare the frequencies among groups, we used the Pearson's chi-square test and the Fisher exact test (when values were less than 5). The association between the variables and anti-*T. gondii* IgG seropositivity was assessed using binary logistic regression analysis with the Enter method. Only statistically significant factors (*P* value <0.05) obtained in the bivariate analysis were included in the binary logistic regression analysis. No analysis of groups within individual variables (multiple comparisons) was performed. A *P*-value < 0.05 was considered statistically significant.

## Ethical aspects

Participants were informed about the aims and methodology of the survey. Enrollment in the study was voluntary. This study was approved by the General Hospital Institutional Review Board of the Secretary of Health in Durango City, Mexico (Approval No. 449/015). Written informed consent was obtained from all participants and from the next of kin for minor participants.

## Results

Most participants were female (65.7%) and were born in the state of Durango (89.8%). Mean age of participants was 39.4±14.0 years (range: 15–93 years). Of the 1544 people surveyed, 173 (11.2%) were positive for anti-*T. gondii* IgG antibodies. Of these 173 seropositive individuals, 82 (47.4%) had anti-*T. gondii* IgG levels between 8 to 99 IU/ml, 15 (8.7%) between 100 to 150 IU/ml, and 76 (43.9%) higher than 150 IU/ml. We were able to test 167 of the 173 anti-*T. gondii* IgG positive sera for anti-*T. gondii* IgM antibodies. Fifty-five (32.9%) of these 167 serum samples were positive for anti-*T. gondii* IgM antibodies. Table 1 shows the sociodemographic and housing data of participants and the seroprevalence of *T. gondii* infection. Bivariate analysis showed that the sociodemographic characteristics gender, residence area, educational level, occupation, and socioeconomic status were significantly (*P*<0.05) associated with *T. gondii* IgG seropositivity, whereas the variables age group, birthplace, and place of residence showed *P* values higher than 0.05 in the bivariate analysis. Concerning housing characteristics, bivariate analysis showed that availability of potable water, type of flooring, form of elimination of excretes, and education of the head of the family were significantly (*P*<0.05) associated with *T. gondii* IgG seropositivity. Overcrowding was not associated with *T. gondii* seropositivity. Bivariate analysis showed that the clinical conditions visual impairment, history of surgery, and hepatitis were negatively associated (*P*<0.05) with *T. gondii* infection. Whereas other clinical variables as presence of any disease, dizziness, frequent abdominal pain or headache, impairments in memory, reflexes, and hearing, and history of blood transfusion, organ transplantation, and lymphadenopathy, showed *P* values higher than 0.05 in the bivariate analysis. Of the obstetric characteristics of women, only the variable of pregnancy history was significantly (*P*<0.05) associated with *T. gondii* IgG seropositivity. Table 2 shows the clinical data of participants and the seroprevalence of *T. gondii* infection.

With respect to behavioral characteristics, bivariate analysis showed many variables associated with *T. gondii* seropositivity. Table 3 shows all behavioral characteristics significantly associated with *T. gondii* seropositivity in the bivariate analysis. The variables domestic travel, consumption of beef, mutton, chicken, turkey or horsemeat, frequency of meat consumption, degree of meat cooking, consumption of dried meat, sausages, and frequency of eating away from home were not associated with *T. gondii* seropositivity. Logistic regression analysis

**Table 1. Socio-demographic and housing characteristics of participants and seroprevalence of *T. gondii* infection.**

| Characteristic | Participants tested No. | Prevalence of *T. gondii* infection | | *P* value |
|---|---|---|---|---|
| | | **No.** | **%** | |
| Age groups (years) | | | | |
| 30 or less | 460 | 39 | 8.5 | 0.07 |
| 31–50 | 746 | 90 | 12.1 | |
| >50 | 338 | 44 | 13.0 | |
| Gender | | | | |
| Male | 530 | 90 | 17.0 | 0.00 |
| Female | 1014 | 83 | 8.2 | |
| Birthplace | | | | |
| Durango State | 1387 | 148 | 10.7 | 0.06 |
| Other Mexican State | 151 | 25 | 16.6 | |
| Abroad | 6 | 0 | 0.0 | |
| Residence place | | | | |
| Durango State | 1532 | 171 | 11.2 | 0.71 |
| Other Mexican State | 11 | 2 | 18.2 | |
| Abroad | 1 | 0 | 0.0 | |
| Residence area | | | | |
| Urban | 1200 | 86 | 7.2 | 0.00 |
| Suburban | 190 | 29 | 15.3 | |
| Rural | 154 | 58 | 37.7 | |
| Educational level | | | | |
| No education | 22 | 8 | 36.4 | 0.00 |
| 1 to 6 years | 248 | 65 | 26.2 | |
| 7–12 years | 773 | 71 | 9.2 | |
| >12 years | 501 | 29 | 5.8 | |
| Occupation | | | | |
| Agriculture | 36 | 6 | 16.7 | 0.00 |
| Housewife | 401 | 53 | 13.2 | |
| Business | 94 | 7 | 7.4 | |
| Construction | 11 | 1 | 9.1 | |
| Employee | 386 | 30 | 7.8 | |
| Student | 97 | 4 | 4.1 | |
| Cattle raising | 4 | 0 | 0.0 | |
| Day laborer | 7 | 1 | 14.3 | |
| Factory worker | 27 | 3 | 11.1 | |
| Professional | 238 | 10 | 4.2 | |
| Miner | 67 | 42 | 62.7 | |
| Sex worker | 27 | 3 | 11.1 | |
| None | 43 | 1 | 2.3 | |
| Other | 106 | 12 | 11.3 | |
| Socio-economic level | | | | |
| Low | 353 | 62 | 17.6 | 0.00 |
| Medium | 1184 | 109 | 9.2 | |
| High | 7 | 2 | 28.6 | |
| Floor at home | | | | |
| Ceramic or wood | 932 | 75 | 8.0 | 0.00 |

(*Continued*)

**Table 1.** (Continued)

| Characteristic | Participants tested No. | Prevalence of *T. gondii* infection | | *P* value |
|---|---|---|---|---|
| | | **No.** | **%** | |
| Concrete | 578 | 86 | 14.9 | |
| Soil | 32 | 12 | 37.5 | |
| Availability of potable water | | | | |
| In the home | 1195 | 86 | 7.2 | 0.00 |
| In the land | 24 | 4 | 16.7 | |
| In the street | 194 | 60 | 30.9 | |
| Toilet facilities | | | | |
| Sewage pipes | 1304 | 92 | 7.1 | 0.00 |
| Latrine or another | 122 | 58 | 47.5 | |
| Crowding at home | | | | |
| No | 579 | 52 | 9.0 | 0.05 |
| Semi-crowded | 538 | 57 | 10.6 | |
| Overcrowded | 277 | 40 | 14.4 | |
| Education of the head of family | | | | |
| Seven years or more | 920 | 60 | 6.5 | 0.00 |
| Four to six years | 360 | 57 | 15.8 | |
| Up to 3 years | 140 | 35 | 25.0 | |

**Table 2. Bivariate analysis of clinical data and infection with *T. gondii* in people with alcohol consumption.**

| Characteristic | Subjects tested No. | Prevalence of *T. gondii* infection | | *P* value |
|---|---|---|---|---|
| | | **No.** | **%** | |
| Clinical status | | | | |
| Healthy | 1093 | 127 | 11.6 | 0.46 |
| Ill | 446 | 46 | 10.3 | |
| Lymphadenopathy ever | | | | |
| Yes | 412 | 40 | 9.7 | 0.24 |
| No | 1126 | 133 | 11.8 | |
| Abdominal pain | | | | |
| Yes | 503 | 49 | 9.7 | 0.20 |
| No | 1039 | 124 | 11.9 | |
| Headache frequently | | | | |
| Yes | 768 | 89 | 11.6 | 0.64 |
| No | 774 | 84 | 10.9 | |
| Memory impairment | | | | |
| Yes | 597 | 60 | 10.1 | 0.29 |
| No | 942 | 111 | 11.8 | |
| Dizziness | | | | |
| Yes | 448 | 55 | 11.3 | 0.97 |
| No | 1053 | 118 | 11.2 | |
| Reflexes impairment | | | | |
| Yes | 257 | 29 | 11.3 | 0.97 |
| No | 1285 | 144 | 11.2 | |

(*Continued*)

**Table 2.** (Continued)

| Characteristic | Subjects tested No. | Prevalence of *T. gondii* infection | | *P* value |
|---|---|---|---|---|
| | | No. | % | |
| Hearing impairment | | | | |
| Yes | 233 | 27 | 11.6 | 0.84 |
| No | 1310 | 146 | 11.1 | |
| Visual impairment | | | | |
| Yes | 577 | 51 | 8.8 | 0.02 |
| No | 965 | 122 | 12.6 | |
| Surgery ever | | | | |
| Yes | 892 | 87 | 9.8 | 0.03 |
| No | 650 | 86 | 13.2 | |
| Transplantation | | | | |
| Yes | 3 | 0 | 0 | 1.00 |
| No | 1538 | 173 | 11.2 | |
| Blood transfusion | | | | |
| Yes | 206 | 25 | 12.1 | 0.65 |
| No | 1335 | 148 | 11.1 | |
| Hepatitis | | | | |
| Yes | 88 | 4 | 4.5 | 0.04 |
| No | 1454 | 169 | 11.6 | |
| Pregnancies | | | | |
| Yes | 850 | 77 | 9.1 | 0.03 |
| No | 150 | 6 | 4.0 | |
| Deliveries | | | | |
| Yes | 609 | 61 | 10.0 | 0.13 |
| No | 239 | 16 | 6.7 | |
| Cesarean sections | | | | |
| Yes | 382 | 37 | 9.7 | 0.57 |
| No | 466 | 40 | 8.6 | |
| Miscarriages | | | | |
| Yes | 284 | 28 | 9.9 | 0.58 |
| No | 562 | 49 | 8.7 | |
| Stillbirths | | | | |
| Yes | 49 | 6 | 12.2 | 0.43 |
| No | 789 | 70 | 8.9 | |

showed that *T. gondii* seropositivity was associated with the following variables: consumption of armadillo meat (OR = 2.33; 95% CI: 1.04–5.22; *P* = 0.03), and the use of latrines for elimination of excretes (OR = 2.27; 95% CI: 1.07–4.80; *P* = 0.03) (Table 4). Other sociodemographic, housing, or behavioral variables included in our study were not associated with *T. gondii* seropositivity by logistic regression analysis.

Our final task was to determine the association between high (>150 IU/ml) anti-*T. gondii* IgG antibody levels and the sociodemographic, housing, and behavioral characteristics of participants. Bivariate analysis showed that several variables are associated (*P*<0.05) with high (>150 IU/ml) anti-*T. gondii* IgG antibody levels. The variables associated with high anti-*T. gondii* IgG antibody levels by bivariate analysis were selected for further analysis by logistic regression analysis (Table 5). This additional analysis showed that high anti-*T. gondii* IgG

**Table 3. Bivariate analysis of selected behavioral factors and infection with *T. gondii* in the population studied.**

| Characteristic | Subjects tested No. | Prevalence of *T. gondii* infection | | P value |
| --- | --- | --- | --- | --- |
| | | No. | % | |
| Cats at home | | | | |
| Yes | 543 | 89 | 16.4 | 0.00 |
| No | 999 | 84 | 8.4 | |
| Cats in the neighborhood | | | | |
| Yes | 1011 | 127 | 12.6 | 0.02 |
| No | 531 | 46 | 8.7 | |
| Cleaning cat excrement | | | | |
| Yes | 400 | 57 | 14.3 | 0.02 |
| No | 1134 | 115 | 10.1 | |
| Birds at home | | | | |
| Yes | 480 | 70 | 14.6 | 0.005 |
| No | 1061 | 103 | 9.7 | |
| Raising farm animals | | | | |
| Yes | 423 | 87 | 20.6 | 0.00 |
| No | 1119 | 86 | 7.7 | |
| Traveled abroad | | | | |
| Yes | 430 | 31 | 7.2 | 0.002 |
| No | 1112 | 142 | 12.8 | |
| Goat meat consumption | | | | |
| Yes | 312 | 71 | 22.8 | 0.00 |
| No | 1227 | 102 | 8.3 | |
| Boar meat consumption | | | | |
| Yes | 167 | 59 | 35.3 | 0.00 |
| No | 1372 | 114 | 8.3 | |
| Pigeon meat consumption | | | | |
| Yes | 129 | 47 | 36.4 | 0.00 |
| No | 1411 | 126 | 8.9 | |
| Duck meat consumption | | | | |
| Yes | 104 | 23 | 22.1 | 0.00 |
| No | 1435 | 150 | 10.5 | |
| Quail meat consumption | | | | |
| Yes | 117 | 36 | 30.8 | 0.00 |
| No | 1423 | 137 | 9.6 | |
| Rabbit meat consumption | | | | |
| Yes | 346 | 62 | 17.9 | 0.00 |
| No | 1194 | 111 | 9.3 | |
| Venison consumption | | | | |
| Yes | 405 | 89 | 22.0 | 0.00 |
| No | 1134 | 83 | 7.3 | |
| Squirrel meat consumption | | | | |
| Yes | 155 | 48 | 31.0 | 0.00 |
| No | 1384 | 125 | 9.0 | |
| Opossum meat consumption | | | | |
| Yes | 60 | 30 | 50.0 | 0.00 |
| No | 1478 | 142 | 9.6 | |

(*Continued*)

**Table 3.** (Continued)

| Characteristic | Subjects tested No. | Prevalence of *T. gondii* infection | | P value |
|---|---|---|---|---|
| | | No. | % | |
| Armadillo meat consumption | | | | |
| Yes | 72 | 40 | 55.6 | 0.00 |
| No | 1467 | 133 | 9.1 | |
| Iguana meat consumption | | | | |
| Yes | 49 | 17 | 34.7 | 0.00 |
| No | 1492 | 156 | 10.5 | |
| Snake meat consumption | | | | |
| Yes | 200 | 36 | 18.0 | 0.001 |
| No | 1341 | 137 | 10.2 | |
| Fish consumption | | | | |
| Yes | 1380 | 163 | 11.8 | 0.03 |
| No | 161 | 10 | 6.2 | |
| Skunk meat consumption | | | | |
| Yes | 22 | 11 | 50.0 | 0.00 |
| No | 1522 | 162 | 10.6 | |
| Chorizo consumption | | | | |
| Yes | 1395 | 164 | 11.8 | 0.04 |
| No | 147 | 9 | 6.1 | |
| Consumption of cow brain | | | | |
| Yes | 287 | 45 | 15.7 | 0.008 |
| No | 1257 | 128 | 10.2 | |
| Liver consumption | | | | |
| Yes | 883 | 118 | 13.4 | 0.00 |
| No | 546 | 34 | 6.2 | |
| Cow raw milk consumption | | | | |
| Yes | 468 | 76 | 16.2 | 0.00 |
| No | 1076 | 97 | 9.0 | |
| Goat raw milk consumption | | | | |
| Yes | 37 | 12 | 32.4 | 0.00 |
| No | 1507 | 161 | 10.7 | |
| Unwashed raw vegetables | | | | |
| Yes | 325 | 69 | 21.2 | 0.00 |
| No | 1219 | 104 | 8.5 | |
| Unwashed raw fruits | | | | |
| Yes | 414 | 81 | 19.6 | 0.00 |
| No | 1128 | 92 | 8.2 | |
| Untreated water | | | | |
| Yes | 564 | 99 | 17.6 | 0.00 |
| No | 976 | 74 | 7.6 | |
| Tobacco consumption | | | | |
| Yes | 727 | 97 | 13.3 | 0.01 |
| No | 817 | 76 | 9.3 | |
| Drug use | | | | |
| Yes | 159 | 27 | 17.0 | 0.01 |
| No | 1383 | 146 | 10.6 | |

(*Continued*)

**Table 3.** (Continued)

| Characteristic | Subjects tested No. | Prevalence of *T. gondii* infection | | *P* value |
|---|---|---|---|---|
| | | No. | % | |
| Sexual promiscuity | | | | |
| Yes | 193 | 39 | 20.2 | 0.00 |
| No | 1346 | 133 | 9.9 | |
| Soil contact | | | | |
| Yes | 899 | 124 | 13.8 | 0.00 |
| No | 642 | 49 | 7.6 | |
| Washing hands before eating | | | | |
| Yes | 1448 | 153 | 10.6 | 0.001 |
| No | 91 | 20 | 22.0 | |

antibodies were associated only with consumption of armadillo meat (OR = 2.25; 95% CI: 1.01–5.02; *P* = 0.04) and crowding at home (OR = 1.63; 95% CI: 1.02–2.61; *P* = 0.03).

## Discussion

The seroprevalence and risk factors associated with *T. gondii* infection in people with alcohol consumption are largely unknown. Consequently, we attempted to determine the rate of *T. gondii* seropositivity and the seroprevalence association with the sociodemographic, clinical, behavioral, and housing characteristics in people with alcohol consumption in Durango, Mexico. The 11.2% seroprevalence of *T. gondii* infection found in this study is higher than the 6.1% seroprevalence of this infection reported in the general population of Durango City, Mexico [20], but is lower than the 23.8% *T. gondii* seroprevalence reported in the general population in rural communities in Durango State, Mexico [22]. It is unclear why alcohol consumers had a different seroprevalence of *T. gondii* infection from those in the general population in urban and rural Durango. In these surveys the same enzyme immunoassay was used. However, differences in the risk factors between the population groups might exist. Seroprevalence of *T. gondii* infection increases with age [21] and is high in rural areas [22]. The mean age of people in the present study (39.4±14.0 years) was comparable to the mean age (37.04±16.1 years) found in the survey in Durango City [21], but lower than that (42.5 ± 17.6 years) found in people in rural Durango [22]. The high seroprevalence of *T. gondii* infection in the subset of people from rural areas observed in our present study might have contributed to an increase in the general seroprevalence. However, the seroprevalence found in people with alcohol consumption is lower than those found in other population groups in our region, including waste pickers (21.1%) [23], schizophrenic patients (20%) [24], inmates (21.1%) [25], or miners (60%) [26]. This relatively low seroprevalence found in our study coincides with the finding of a study of the Finnish general population, where researchers found that *T. gondii* seroprevalence was not associated with alcohol use disorders [28]. It is unknown whether *T. gondii*-induced behavioral changes might lead to alcohol consumption or whether alcohol consumption may lead to risky behavior for *T. gondii* infection. This study was performed over a 4-year period; however, this fact does not affect the interpretation of the results as the immunoassays were the same and no change in the seroprevalence of *T. gondii* infection in Durango during the last 10 years has been reported.

We sought to determine the risk factors associated with *T. gondii* seropositivity and serointensity. Logistic regression analysis of sociodemographic, housing, and behavioral characteristics of people with alcohol consumption showed that *T. gondii* seropositivity was associated

**Table 4. Multivariate analysis of selected characteristics of people with alcohol consumption and their association with *T. gondii* infection.**

| Characteristic | Odds ratio | 95% confidence interval | *P* value |
|---|---|---|---|
| Gender (male) | 1.10 | 0.74–1.92 | 0.45 |
| Residence area (area) | 1.70 | 0.37–7.98 | 0.48 |
| Educational level (no education) | 1.00 | 0.24–4.21 | 0.97 |
| Occupation (none) | 0.25 | 0.03–1.98 | 0.19 |
| Socioeconomic level (low) | 1.44 | 0.91–2.26 | 0.11 |
| Cats at home (yes) | 1.48 | 0.92–2.36 | 0.09 |
| Cats in the neighborhood (yes) | 0.83 | 0.52–1.34 | 0.45 |
| Cleaning cat excrement (yes) | 0.90 | 0.55–1.47 | 0.69 |
| Birds at home (yes) | 1.11 | 0.71–1.75 | 0.63 |
| Raising farm animals (yes) | 1.51 | 0.94–2.43 | 0.08 |
| Traveled abroad (yes) | 0.68 | 0.4–1.16 | 0.16 |
| Goat meat consumption (yes) | 0.95 | 0.55–1.66 | 0.87 |
| Boar meat consumption (yes) | 1.44 | 0.73–2.84 | 0.28 |
| Pigeon meat consumption (yes) | 1.08 | 0.52.31 | 0.84 |
| Duck meat consumption (yes) | 1.24 | 0.56–2.74 | 0.59 |
| Quail meat consumption (yes) | 1.31 | 0.622.74 | 0.47 |
| Rabbit meat consumption (yes) | 0.92 | 0.52–1.62 | 0.77 |
| Venison consumption (yes) | 1.41 | 0.82–2.43 | 0.21 |
| Squirrel meat consumption (yes) | 1.28 | 0.65–2.49 | 0.46 |
| Opossum meat consumption (yes) | 0.95 | 0.37–2.45 | 0.93 |
| Armadillo meat consumption (yes) | 2.33 | 1.04–5.22 | 0.03 |
| Iguana meat consumption (yes) | 1.56 | 0.63–3.87 | 0.33 |
| Snake meat consumption (yes) | 0.64 | 0.34–1.22 | 0.18 |
| Fish consumption (yes) | 1.21 | 0.56–2.60 | 0.62 |
| Skunk meat consumption (yes) | 1.09 | 0.27–4.42 | 0.89 |
| Chorizo consumption (yes) | 1.42 | 0.58–3.48 | 0.43 |
| Consumption of cow brain (yes) | 0.95 | 0.55–1.64 | 0.86 |
| Liver consumption (yes) | 1.37 | 0.84–2.25 | 0.20 |
| Cow raw milk consumption (yes) | 0.89 | 0.56–143 | 0.65 |
| Goat raw milk consumption (yes) | 0.47 | 0.14–1.54 | 0.21 |
| Unwashed raw vegetables (yes) | 1.46 | 0.76–2.79 | 0.24 |
| Unwashed raw fruits (yes) | 0.69 | 0.35–1.35 | 0.28 |
| Untreated water (yes) | 1.40 | 0.87–2.23 | 0.15 |
| Tobacco consumption (yes) | 1.21 | 0.79–1.85 | 0.35 |
| Drug use (yes) | 0.82 | 0.43–1.57 | 0.55 |
| Sexual promiscuity (yes) | 1.07 | 0.6–1.89 | 0.80 |
| Soil contact (yes) | 0.99 | 0.62–1.56 | 0.96 |
| Washing hands before eating (yes) | 1.02 | 0.46–2.23 | 0.95 |
| Floor at home (soil) | 0.78 | 0.25–2.42 | 0.66 |
| Availability of potable water (in the street) | 1.69 | 0.92–3.10 | 0.08 |
| Toilet facilities (latrine) | 2.27 | 1.07–4.80 | 0.03 |
| Education of the head of family (up to 3 years) | 1.26 | 0.68–2.33 | 0.44 |

with consumption of armadillo meat, and the use of latrines for elimination of excretes. In an epidemiological study in elderly people in Durango, Mexico, we found that *T. gondii* seropositivity was associated with consumption of armadillo meat [29]. Thus, we demonstrate this

**Table 5. Multivariate analysis of selected characteristics of people with alcohol consumption and their association with high (>150 IU/ml) anti-*T. gondii* IgG antibodies.**

| Characteristic | Odds ratio | 95% confidence interval | *P* value |
|---|---|---|---|
| Age (more than 50 years) | 1.07 | 0.63–1.84 | 0.78 |
| Gender (male) | 1.2 | 0.75–1.90 | 0.43 |
| Residence area (rural) | 1.25 | 0.59–2.67 | 0.55 |
| Educational level (no education) | 0.99 | 0.22–4.31 | 0.99 |
| Occupation (none) | 0.24 | 0.03–1.86 | 0.17 |
| Socioeconomic level (low) | 1.31 | 0.84–2.06 | 0.22 |
| Cats at home (yes) | 1.49 | 0.98–2.25 | 0.05 |
| Raising farm animals (yes) | 1.48 | 0.94–2.33 | 0.08 |
| Traveled abroad (yes) | 0.69 | 0.41–1.18 | 0.18 |
| Goat meat consumption (yes) | 0.9 | 0.52–1.57 | 0.73 |
| Boar meat consumption (yes) | 1.34 | 0.68–2.64 | 0.38 |
| Pigeon meat consumption (yes) | 1.04 | 0.49–2.20 | 0.91 |
| Duck meat consumption (yes) | 1.16 | 0.52–2.58 | 0.7 |
| Quail meat consumption (yes) | 1.32 | 0.63–2.75 | 0.44 |
| Rabbit meat consumption (yes) | 0.92 | 0.52–1.61 | 0.78 |
| Venison consumption (yes) | 1.48 | 0.86–2.54 | 0.14 |
| Squirrel meat consumption (yes) | 1.32 | 0.68–2.55 | 0.4 |
| Opossum meat consumption (yes) | 0.83 | 0.33–2.07 | 0.7 |
| Armadillo meat consumption (yes) | 2.25 | 1.01–5.02 | 0.04 |
| Iguana meat consumption (yes) | 1.48 | 0.60–3.64 | 0.39 |
| Snake meat consumption (yes) | 0.64 | 0.34–1.20 | 0.16 |
| Skunk meat consumption (yes) | 1.12 | 0.29–4.36 | 0.86 |
| Chorizo consumption (yes) | 1.45 | 0.59–3.58 | 0.4 |
| Liver consumption (yes) | 1.39 | 0.86–2.25 | 0.17 |
| Cow raw milk consumption (yes) | 0.74 | 0.46–1.18 | 0.21 |
| Unwashed raw vegetables (yes) | 1.44 | 0.76–2.74 | 0.25 |
| Unwashed raw fruits (yes) | 0.75 | 0.39–1.46 | 0.4 |
| Untreated water (yes) | 1.39 | 0.87–2.21 | 0.15 |
| Washing hands before eating (yes) | 1.06 | 0.49–2.30 | 0.86 |
| Floor at home (soil) | 0.64 | 0.21–1.97 | 0.44 |
| Availability of potable water (in the street) | 1.72 | 0.93–3.19 | 0.08 |
| Toilet facilities (latrine) | 1.94 | 0.82–4.59 | 0.12 |
| Crowding (yes) | 1.63 | 1.02–2.61 | 0.03 |
| Education of the head of the family (up to 3 years) | 1.23 | 0.67–2.24 | 0.5 |

association for the second time in Mexico. Serological evidence of *T. gondii* infection in armadillos has been established in Brazil [30, 31]. However, to the best of our knowledge, no study about *T. gondii* infection in armadillos in Mexico has been reported. Our conclusions coincide with a previous report in Mexico examining the association between *T. gondii* seropositivity and the use of latrines. This association was noted in an epidemiological study of pregnant women in the central Mexican city of Aguascalientes [32]. Additionally, high anti-*T. gondii* IgG antibodies were associated with consumption of armadillo meat and crowding at home. Incidence of toxoplasmosis increases with crowding and poor sanitary habits [33]. The association between high antibody levels and crowding found in the present study is in line with a previous report of an association between seroreactivity (IgG and IgM) to *T. gondii* and crowding in the US [34].

As to clinical characteristics, the negative association between *T. gondii* seropositivity and visual impairment, history of surgery, and hepatitis suggests that *T. gondii* infection did not play an important role in these clinical factors in the study population. In the women studied, the higher frequency of *T. gondii* seropositivity in women with pregnancies than in women without pregnancies may be due to a higher age in women with pregnancies than in those without pregnancies.

The current study has the following limitations: 1) Although the sample size was large, we studied subjects in only one Mexican state, and the results of this study cannot be extrapolated to other Mexican states; and 2) the quantity and frequency of alcohol intake was not correlated with *T. gondii* exposure. Epidemiological studies to evaluate the link between seropositivity and serointensity of *T. gondii* infection and alcohol consumption including several Mexican states and quantification of alcohol intake are needed.

## Conclusions

This is the first study about the epidemiology of *T. gondii* exposure in alcohol consumers. We found a different *T. gondii* seroprevalence in alcohol consumers from those previously found in population groups in the region. This is the first study to show the association between high anti-*T. gondii* antibodies and overcrowding in Mexico, and the second study on the association between *T. gondii* infection and consumption of armadillo meat and the use of latrines in this country. We believe the epidemiology of *T. gondii* infection in people with alcohol consumption merits further investigation. We further believe identification of factors associated with *T. gondii* seropositivity in this study may help inform and contribute to the effective planning of prevention and control measures against infection with *T. gondii* and its sequelae.

## Supporting information

**S1 File.**
(DOCX)

**S2 File.**
(DOCX)

## Acknowledgments

Tolliver Cleveland Callison IV edited the manuscript.

## Author Contributions

**Conceptualization:** Elizabeth Rábago-Sánchez, Cosme Alvarado-Esquivel.

**Data curation:** Sergio Estrada-Martinez, Alma Rosa Pérez-Álamos, Melina Ibarra-Segovia, Isabel Beristaín-Garcia, Agar Ramos-Nevárez, Leandro Saenz-Soto, Carlos Alberto Guido-Arreola, Cosme Alvarado-Esquivel.

**Formal analysis:** Sergio Estrada-Martinez, Alma Rosa Pérez-Álamos, Isabel Beristaín-Garcia, Agar Ramos-Nevárez, Leandro Saenz-Soto, Elizabeth Rábago-Sánchez, Cosme Alvarado-Esquivel.

**Funding acquisition:** Cosme Alvarado-Esquivel.

**Investigation:** Melina Ibarra-Segovia, Isabel Beristaín-Garcia, Agar Ramos-Nevárez, Leandro Saenz-Soto, Carlos Alberto Guido-Arreola, Cosme Alvarado-Esquivel.

**Methodology:** Sergio Estrada-Martinez, Elizabeth Rábago-Sánchez, Cosme Alvarado-Esquivel.

**Project administration:** Cosme Alvarado-Esquivel.

**Software:** Sergio Estrada-Martinez, Alma Rosa Pérez-Álamos.

**Writing – original draft:** Cosme Alvarado-Esquivel.

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
