## [Decision Letter · Decision Letter 0]

23 Nov 2020

PONE-D-20-27939

Seroepidemiology of Toxoplasma gondii infection in people with alcohol consumption

PLOS ONE

Dear Dr. Cosme Alvarado-Esquivel

Thank you for submitting your manuscript to PLOS ONE. After careful consideration, we feel that it has merit but does not fully meet PLOS ONE’s publication criteria as it currently stands. Therefore, we invite you to submit a revised version of the manuscript that addresses the points raised during the review process. I have reviewed both reviewer comments, one of them was "reject" and the other one was Minor revision. Thus, I will give you another chance to address the comments carefully. 

We look forward to receiving your revised manuscript.

Kind regards,

Gheyath K. Nasrallah, PhD

Academic Editor

PLOS ONE

Journal Requirements:

2.We suggest you thoroughly copyedit your manuscript for language usage, spelling, and grammar. If you do not know anyone who can help you do this, you may wish to consider employing a professional scientific editing service.  

4. In the Methods, please discuss whether and how the questionnaire was validated and/or pre-tested. If this did not occur, please provide the rationale for not doing so.

5. Please clearly define alcohol consumption and specifically outline the criteria that were used to determine consumption. The statement "...at least one drink a month..." is non-specific as different drinks have varying concentrations of alcohol.

6. In statistical methods, please clarify whether you corrected for multiple comparisons.

Reviewers' comments:

Reviewer's Responses to Questions

**Comments to the Author**

1. Is the manuscript technically sound, and do the data support the conclusions?

Reviewer #1: No

Reviewer #2: Partly

2. Has the statistical analysis been performed appropriately and rigorously? 

Reviewer #1: Yes

Reviewer #2: Yes

3. Have the authors made all data underlying the findings in their manuscript fully available?

Reviewer #1: Yes

Reviewer #2: No

4. Is the manuscript presented in an intelligible fashion and written in standard English?

Reviewer #1: Yes

Reviewer #2: Yes

5. Review Comments to the Author

Reviewer #1: The authors evaluated the " Seroepidemiology of Toxoplasma gondii infection in people with alcohol consumption". After close review I have recommended that the manuscript cannot be accepted for publication. Please see some major concerns as below:

Add the location of study in the title

Keywords must be different from title words.

The rational of the study is not clearly understood.

About the inclusion criteria, what about other criteria such as immune deficiency diseases, chronic diseases, etc?????

What about the exclusion criteria?

The main disadvantage of this study is the lack of a control group.

Considering serological tests, what about the range of equivocal results and index values for IgG and IgM?

Considering low specificity of IgM and IgG in the ELISA test, it is better that the authors perform the high level tests such as Elecsys Toxo IgM assay, PCR assay, …..

Add ethical statement number

Reviewer #2: Summary:

The authors studied the seroprevalence of toxoplasma in people with alcohol consumption in Durango in Mexico. They also reported the associated factors with being seropositive. The study included 1544 participants during the period 2014 to 2018 and found 11.2 % prevalence. The authors used bivariate analysis to compare the Prevalence Ratio among participants with different sociodemographic and clinical characteristics where they identified the characteristics associated with higher (or lower) prevalence ratio. The authors then included variables with statistical significance in a logistic regression model where they compared the Odd Ratio and identified consumption of armadillo meat, the use of latrine for elimination of excretes, and crowding at home as associated factors with being positive.

The authors addressed an interesting topic in a presentable fashion. They studied a wide range of characteristics which were quite compressive. The statistical methods were appropriately used to present the results.

The authors didn’t give clear description of the study setting, the sampling frame and the sampling technique.

Issues:

1- Authors might need to give a bit of context for the study settings including if it was conducted in Durango state or Durango city. It was not clear to the reviewer what area does this study cover. It was not clear as well where the recruitment of participant in the study was conducted (community, health facilities, ….???).

2- Would you explain the number mentioned in line 134 (336,606) as the size of population from which the sample was selected?

3- What was the sampling technique employed in this study? How the participants in the study were selected and enrolled?

4- Would you please highlight if the study duration of 4 years (2014 – 2018) would impact the interpretation of the results?

5- In table 5, the authors presented the logistic regression results of high IgG and some characteristics. The bivariate analysis of this characteristics was not reported in the manuscript.

6- The authors didn’t consider a P value of exactly 0.05 as significant, which is debatable.

7- In line 172, the authors stated that (crowding was not associated with seropositivity). However, it was mentioned that crowding was associated with (high) antibodies. The authors need to define the term (seropositivity) and make clear distinction between IgG, IgM and both IgG/IgM positive cases. Moreover, the clinical and statistical significance of the levels of IgG need to be explained.

8- It was mentioned in line 111 and 112 that “traveling, frequency of eating away from home (in restaurants or fast food outlets)” were studied as an associated factors. In the results section in lines 184 to 188, authors reported that these factors were not found to be associated with seropositivity. However, the reviewer failed to find these factors among the results tables.

9- In lines 173, 174 and 175, the authors mentioned that visual impairment, history of surgery, and hepatitis were associated with IgG seropositivity. This should be corrected to (negatively associated) as mentioned in line 36.

10- In line 33, the sentence starting (fitty-five …..) is confusing. Please consider mention first that 167 of the IgG positives were also tested for IgM and 55 of them were positive.

11- In line 29, please replace the word (detected) by another word (e.g. studies, measured) to avoid confusion.

12- Please consider how this study will inform the prevention and control measures.

6. PLOS authors have the option to publish the peer review history of their article (what does this mean?). If published, this will include your full peer review and any attached files.

Reviewer #1: No

Reviewer #2: **Yes: **Sayed Himatt

---

## [Author Response · Author response to Decision Letter 0]

14 Dec 2020

Durango, Dgo. Mexico. December 14, 2020.

Dear Editor,

Please find attached a revised version of our manuscript that has been modified according to the reviewers’ comments. In addition, please find below our response to each of the reviewers’ comments on a point-by-point basis.

We appreciate the valuable comments of the reviewers and we hope the revised manuscript may have more success for publication in the journal Plos One.

Kind regards,

Dr. Cosme Alvarado-Esquivel.

Laboratorio de Investigación Biomédica

Facultad de Medicina y Nutrición

Avenida Universidad S/N.

34000 Durango, Dgo. Mexico.

Tel/Fax.: 0052 618 8 271200

Email: alvaradocosme@yahoo.com

 

RESPONSE TO THE REVIEWERS’ COMMENTS

The manuscript was modified according to the PLOS ONE’s style requirements. 

2.We suggest you thoroughly copyedit your manuscript for language usage, spelling, and grammar. If you do not know anyone who can help you do this, you may wish to consider employing a professional scientific editing service. 

English was revised. The name of the American who review the manuscript was written in the Acknowledgements section. Changes were highlighted in the manuscript. A file (“Supporting information”) with the changes marked in green color was included. A clean copy of the edited manuscript (“Manuscript”) was included. 

A questionnaire was included in both the original language and English.

4. In the Methods, please discuss whether and how the questionnaire was validated and/or pre-tested. If this did not occur, please provide the rationale for not doing so.

There was no need to perform a validation of the questionnaire because it has been used in a number of epidemiological studies in the same language in the same country. The questionnaire was not used as an evaluation scale but for recording simple variables as age, gender, etc. Items of housing variables were previously validated. This information was added to the Methods section (lines 108-113). 

5. Please clearly define alcohol consumption and specifically outline the criteria that were used to determine consumption. The statement "...at least one drink a month..." is non-specific as different drinks have varying concentrations of alcohol.

Further information about the definition of alcohol consumption was added to the Methods section (lines 91-96).

6. In statistical methods, please clarify whether you corrected for multiple comparisons.

No analysis of groups within individual variables (multiple comparisons) was performed (lines 163-165). 

Thank you for your valuable comments for improving our manuscript.

Reviewers' comments:

Reviewer's Responses to Questions

Comments to the Author

1. Is the manuscript technically sound, and do the data support the conclusions?

Reviewer #1: No

Reviewer #2: Partly

2. Has the statistical analysis been performed appropriately and rigorously?

Reviewer #1: Yes

Reviewer #2: Yes

3. Have the authors made all data underlying the findings in their manuscript fully available?

Reviewer #1: Yes

Reviewer #2: No

4. Is the manuscript presented in an intelligible fashion and written in standard English?

Reviewer #1: Yes

Reviewer #2: Yes

5. Review Comments to the Author

Reviewer #1: 

1. Add the location of study in the title

The location of the study was added to the Title.

2. Keywords must be different from title words.

Changed. All keywords are now different from the Title words. 

3. The rational of the study is not clearly understood.

The rationale section was rewritten to make it clearer (lines 68-80).

4. About the inclusion criteria, what about other criteria such as immune deficiency diseases, chronic diseases, etc?????

The presence of any disease was not a restrictive criterium for inclusion (lines 96-98). 

5. What about the exclusion criteria?

Exclusion criteria were added (lines 98-99).

6. The main disadvantage of this study is the lack of a control group.

This is not a case control study; it was not aimed to determine an association between T. gondii infection and alcohol consumption. This study is a cross-sectional study aimed to determine the prevalence of and factors associated with T. gondii infection in people with alcohol consumption. 

7. Considering serological tests, what about the range of equivocal results and index values for IgG and IgM?

There were no equivocal results. Information about index values for IgG and IgM was added (lines 137-140).

8. Considering low specificity of IgM and IgG in the ELISA test, it is better that the authors perform the high level tests such as Elecsys Toxo IgM assay, PCR assay, …..

This study was aimed to determine the seroepidemiology of T. gondii infection and this aim can only be reached by determining the anti-T. gondii IgG and IgM antibodies. These infection markers are well accepted markers used for epidemiological studies. 

9. Add ethical statement number

Added (lines 170-171).

Thank you for your valuable comments for improving our manuscript.

Reviewer #2: 

1- Authors might need to give a bit of context for the study settings including if it was conducted in Durango state or Durango city. It was not clear to the reviewer what area does this study cover. It was not clear as well where the recruitment of participant in the study was conducted (community, health facilities, ….???).

Information about the study settings was added (lines 99-102).

2- Would you explain the number mentioned in line 134 (336,606) as the size of population from which the sample was selected?

An explanation about the size of population from which the sample was selected was added (lines 153-158).

3- What was the sampling technique employed in this study? How the participants in the study were selected and enrolled?

Information about sampling was added (lines 101-103). 

4- Would you please highlight if the study duration of 4 years (2014 – 2018) would impact the interpretation of the results?

Information about a lack of impact of the duration of the study on the interpretations of the results was added to the Discussion section (lines 246-249).

5- In table 5, the authors presented the logistic regression results of high IgG and some characteristics. The bivariate analysis of this characteristics was not reported in the manuscript.

Information about results of bivariate analysis of high IgG antibody levels and the characteristics of participants was added (lines 211-219).

6- The authors didn’t consider a P value of exactly 0.05 as significant, which is debatable.

We used a P<0.05 as statistically significant just to be more stringent in determining associations between T. gondii infection and other variables. 

7- In line 172, the authors stated that (crowding was not associated with seropositivity). However, it was mentioned that crowding was associated with (high) antibodies. The authors need to define the term (seropositivity) and make clear distinction between IgG, IgM and both IgG/IgM positive cases. Moreover, the clinical and statistical significance of the levels of IgG need to be explained.

An explanation about the terms seropositivity, serointensity and high antibody levels were added (lines 141-146).

8- It was mentioned in line 111 and 112 that “traveling, frequency of eating away from home (in restaurants or fast food outlets)” were studied as an associated factors. In the results section in lines 184 to 188, authors reported that these factors were not found to be associated with seropositivity. However, the reviewer failed to find these factors among the results tables.

More information about “traveling” was added (line 120). The words “traveled abroad” were included in the Tables. The words “domestic travel” are mentioned in line 203. 

The variable “frequency of eating away from home” is mentioned in line 205. 

9- In lines 173, 174 and 175, the authors mentioned that visual impairment, history of surgery, and hepatitis were associated with IgG seropositivity. This should be corrected to (negatively associated) as mentioned in line 36.

Corrected (lines 192-193).

10- In line 33, the sentence starting (fitty-five …..) is confusing. Please consider mention first that 167 of the IgG positives were also tested for IgM and 55 of them were positive.

The sentence was modified (lines 32-34).

11- In line 29, please replace the word (detected) by another word (e.g. studies, measured) to avoid confusion.

Changed by “searched” (lines 27-29).

12- Please consider how this study will inform the prevention and control measures.

Added (lines 288-291).

Thank you for your valuable comments for improving our manuscript.

---

## [Decision Letter · Decision Letter 1]

6 Jan 2021

Seroepidemiology of Toxoplasma gondii infection in people with alcohol consumption in Durango, Mexico

PONE-D-20-27939R1

Dear Dr. Cosme Alvarado-Esquivel,

We’re pleased to inform you that your manuscript has been judged scientifically suitable for publication and will be formally accepted for publication once it meets all outstanding technical requirements.

Kind regards,

Gheyath K. Nasrallah, PhD

Academic Editor

PLOS ONE

Additional Editor Comments (optional):

Reviewers' comments:

Reviewer's Responses to Questions

**Comments to the Author**

1. If the authors have adequately addressed your comments raised in a previous round of review and you feel that this manuscript is now acceptable for publication, you may indicate that here to bypass the “Comments to the Author” section, enter your conflict of interest statement in the “Confidential to Editor” section, and submit your "Accept" recommendation.

Reviewer #2: All comments have been addressed

2. Is the manuscript technically sound, and do the data support the conclusions?

Reviewer #2: Yes

3. Has the statistical analysis been performed appropriately and rigorously? 

Reviewer #2: Yes

4. Have the authors made all data underlying the findings in their manuscript fully available?

Reviewer #2: Yes

5. Is the manuscript presented in an intelligible fashion and written in standard English?

Reviewer #2: Yes

6. Review Comments to the Author

Reviewer #2: (No Response)

7. PLOS authors have the option to publish the peer review history of their article (what does this mean?). If published, this will include your full peer review and any attached files.

Reviewer #2: **Yes: **Sayed Himatt

---

## [Editor Report · Acceptance letter]

11 Jan 2021

PONE-D-20-27939R1 

Seroepidemiology of *Toxoplasma gondii* infection in people with alcohol consumption in Durango, Mexico 

Dear Dr. Alvarado-Esquivel:

I'm pleased to inform you that your manuscript has been deemed suitable for publication in PLOS ONE. Congratulations! Your manuscript is now with our production department. 

Kind regards, 

on behalf of

Dr. Gheyath K. Nasrallah 

Academic Editor

PLOS ONE